# Spiking Meets Attention: Efficient Remote Sensing Image Super-Resolution with Attention Spiking Neural Networks

Yi Xiao[1]    Qiangqiang Yuan[2]*    Kui Jiang[3]    Wenke Huang[2]    Qiang Zhang[4]
Tingting Zheng[3]    Chia-Wen Lin[5]    Liangpei Zhang[2]

[1]School of Computer and Artificial Intelligence, Zhengzhou University
[2]Wuhan University  [3]Harbin Institution of Technology  [4]Dalian Maritime University
[5]National Tsinghua University
yixiao@zzu.edu.cn

## Abstract

Spiking neural networks (SNNs) are emerging as a promising alternative to traditional artificial neural networks (ANNs), offering biological plausibility and energy efficiency. Despite these merits, SNNs are frequently hampered by limited capacity and insufficient representation power, yet remain underexplored in remote sensing image (RSI) super-resolution (SR) tasks. In this paper, we first observe that spiking signals exhibit drastic intensity variations across diverse textures, highlighting an active learning state of the neurons. This observation motivates us to apply SNNs for efficient SR of RSIs. Inspired by the success of attention mechanisms in representing salient information, we devise the spiking attention block (SAB), a concise yet effective component that optimizes membrane potentials through inferred attention weights, which, in turn, regulates spiking activity for superior feature representation. Our key contributions include: 1) we bridge the independent modulation between temporal and channel dimensions, facilitating joint feature correlation learning, and 2) we access the global self-similar patterns in large-scale remote sensing scenarios to infer spatial attention weights, incorporating effective priors for realistic and faithful reconstruction. Building upon SAB, we proposed SpikeSR, which achieves state-of-the-art performance across various remote sensing benchmarks such as AID, DOTA, and DIOR, while maintaining high computational efficiency. Code of SpikeSR will be available at https://github.com/XY-boy/SpikeSR.

High-resolution remote sensing images (RSIs) contain fine-grained object structures and textures, which are critical for accurate interpretation in downstream tasks [42, 11, 6]. However, limited by the intrinsic resolution of airborne sensors, RSI can merely capture partial spatial details, resulting in suboptimal scene representation and visual quality. Image super-resolution (SR) aims to alleviate this problem by reconstructing high-resolution (HR) images from low-resolution (LR) observations [53, 60]. Despite this, SR remains a challenging ill-posed issue, as a degraded input may correspond to multiple plausible outputs.

Early efforts rely on hand-crafted priors to tame the ill-posedness, *e.g.*, nonlocal mean [67, 7] and gradient profile [48], but they are often trapped in limited performance and scalability. Recent advances in artificial neural networks (ANNs), *e.g.*, CNNs and Transformers, have witnessed remarkable progress in SR with large-capacity models [49, 50, 3, 4, 71]. However, they often come with a

---

*Corresponding Author.

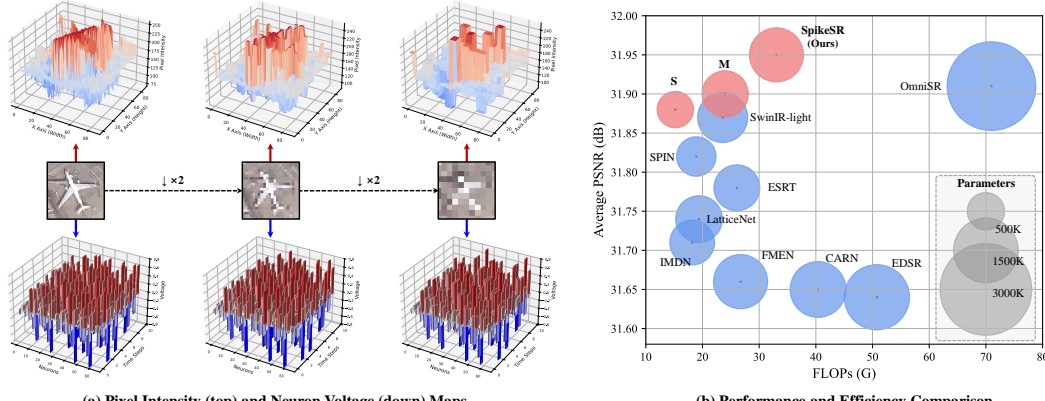

(a) Pixel Intensity (top) and Neuron Voltage (down) Maps

(b) Performance and Efficiency Comparison

Figure 1: (a) The visualization of pixel intensity and neuron voltage in images under various degradation factors reveals important insights. The pixel intensity map illustrates that the high-frequency components of the image tend to be smooth, indicating a reduction in sharp details during progressive downsampling. Neuron intensity maps, derived from a LIF model [37, 14], show that high-frequency details persist with drastic fluctuations, suggesting that the neurons remain in an active state. (b) FLOPs and PSNR performance comparison. The circle sizes represent the number of parameters. Our SpikeSR outperforms SOTA efficient SR methods with high efficiency. PSNR results are averaged on the AID, DOTA, and DIOR datasets.

trade-off of increased computational overhead and growing storage costs, making them less efficient in practical scenarios, particularly when reconstructing large-scale RSIs.

More recently, brain-inspired spiking neural networks (SNNs), as the third generation of neural networks, have emerged as a promising alternative for energy-efficient intelligence [25, 59, 40]. Different from ANNs that encode features as continuous values, SNNs can emulate biological communication with discrete spiking signals and propagate them by neurons, thus enjoying lower power consumption. As depicted in Fig. 1(a), our experiments reveal a novel finding that spiking neurons maintain an active learning state across LR RSIs, even in severely damaged textures. Specifically, we observed that degraded RSIs exhibit smoothed pixel intensities and obscured sharp details, posing a significant challenge to characterize high-frequency representations. In contrast, spiking signals retain drastic responses and pronounced spike rates, highlighting that neurons remain in an active learning state. This naturally arises a question: *Can SNNs leverage their inherent properties to handle image degradation for efficient yet high-quality RSI SR?*

In fact, to effectively grasp complex and diverse spatial details in RSIs, the network must possess adequate capacity and representation power. Unfortunately, there are two critical challenges when adapting SNNs for SR tasks. Firstly, **spiking activity in SNNs inevitably causes pixel-wise information loss,** which hampers the representation capacity of SNNs, especially when the network deepens. This stems from the discrete nature of binary spiking signals, leading to undesirable spiking degradation problems [61, 15]. Secondly, **SNNs remain constrained by suboptimal membrane potential dynamics,** restricting effective exploration of global context during spiking communications. This necessitates a customized strategy to optimize membrane potentials, but is barely explored before.

To address these limitations, we propose SpikeSR, an SNN-based framework inspired by human visual attention mechanisms, which can actively represent image degradation and modulate synaptic weights to focus on salient regions, which, in turn, regulate the spiking activity for improved capacity and representation power. Specifically, SpikeSR employs a concise yet effective spiking attention block (SAB) to optimize feature emphasis through spiking response dynamics, which integrates three key innovations: 1) the combination of CNN and SNN layers to mitigate information loss induced by discrete spiking activity; 2) introducing hybrid dimension attention (HDA) to recalibrate spiking response across both temporal and channel dimensions, facilitating a joint feature correlations learning; 3) accessing global self-similarity patterns in RSIs to infer spatial attention weights, incorporating effective priors for realistic and faithful reconstruction. Compared to state-of-the-art (SOTA) ANN-based efficient SR models, our SpikeSR demonstrates lower model complexity and superior performance, as shown in Fig. 1(b).

Our contributions are summarized as follows:

- We pioneer an attention spiking neural network for efficient SR of RSIs, providing a new perspective on developing efficient models in large-scale Earth observation scenarios.

- We devise a concise yet effective SAB, which mitigates the information loss and regulates membrane potentials of spiking activity for improved representation of SNNs.

- Extensive experimental results on various remote sensing datasets demonstrate that our SpikeSR achieves competitive SR performance against SOTA ANN-based methods.

# 1 Related Work

**Deep Networks for SR**. Inspired by the pioneering SRCNN [10], CNN-based SR methods have achieved remarkable progress, dominating the field for years. They mainly elaborated on the network design to tame the ill-posedness, with notable advances in residual connections [21, 32] and attention mechanisms [69, 39]. However, these methods often suffer from high computational complexity, *e.g.*, exhaustive non-local modeling [28, 38], making them less efficient in large-scale RSIs.

Recently, transformer-based SR models have demonstrated impressive performance, benefiting from their ability to model long-range dependencies. IPT [2] first introduces Transformers in SR field, but requires massive parameters and laborious pre-training processes. SwinIR [31] effectively reduces the model size by partitioning the image into smaller windows when applying multi-head attention mechanisms, while maintaining favorable performance. Against transformer, Mamba-based SR methods achieve comparable global model capacity with linear complexity [18, 57, 17]. Despite these advancements, advanced SR models are often trapped by rising computational overhead and growing storage costs, posing significant concerns in real-world applications, particularly in remote sensing scenarios.

**Efficient SR Models.** To reduce computational budget, CARN [1] utilizes grouped convolutions and a cascading mechanism to improve the residual architecture. IMDN [20] progressively distills useful information during feature extraction and applies network pruning to further decrease complexity. FMEN [12] optimizes residual modules to accelerate inference. In Transformer-based SR methods, SPIN [66] enhances long-range modeling by combining self-attention with pixel clustering, facilitating interactions between superpixels. HiT-SR [68] expands the self-attention receptive field by applying different window sizes of hierarchal layers. Despite these successes, there is still room to further boost SR performance. Moreover, the potential of energy-efficient SNNs for SR tasks remains largely unexplored.

**Spiking Neural Networks.** Recent advances in neuromorphic computing have shown the great potential of SNNs in computational efficiency and power as CNNs [63, 64]. Currently, SNNs have been successfully applied to various tasks, such as image classification [25, 44], object detection and tracking [22, 59], optical flow estimation [26, 40], *etc*.

A common solution to build SNNs is converting pre-trained ANN models [8, 62]. Li *et al.* [30] proposed a layer-wise calibration to minimize activation mismatch during conversion. Ding *et al.* [9] replaced ReLU with the rate norm layer, enabling direct conversion from a trained ANN to an SNN. Stockl *et al.* [46] used time-varying multi-bit spikes to better approximate activation functions. However, conversion-based methods face accuracy gaps and high latency due to extensive time-step simulations, resulting in increased latency and energy consumption.

An alternative involves using agent gradient functions for continuous relaxation of non-smooth spike activities, enabling direct training via backpropagation through time. Lee *et al.* [27] treated membrane potential as a signal to overcome discontinuities, enabling direct training from spikes. Wang *et al.* [54] introduced an iterative LIF model and proposed spatiotemporal backpropagation based on approximate peak activity derivatives. Later, Zheng *et al.* [70] proposed temporal delay batch normalization, which significantly enhanced the depth of SNNs. To bridge the performance gap between ANNs and SNNs, some methods borrowed insights from CNNs, applying residual learning [15, 19] and attention mechanisms [65, 41] to SNNs. Nonetheless, there has been limited exploration of pixel-level regression tasks, such as SR.

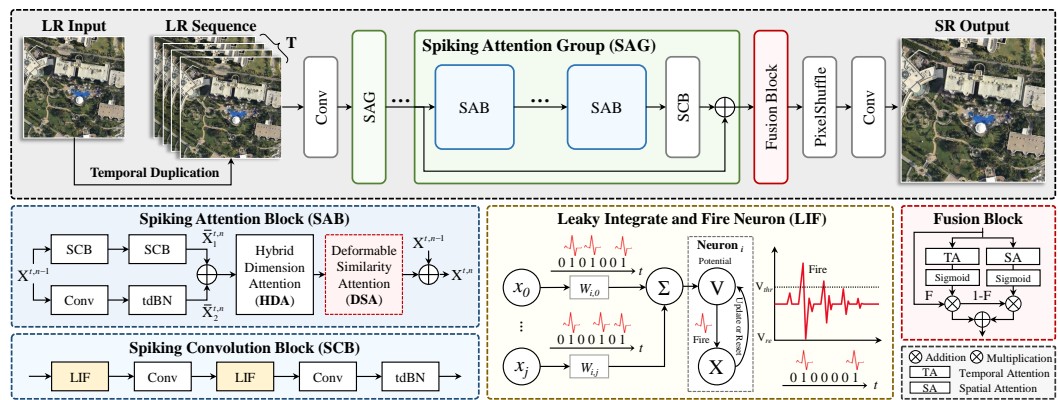

Figure 2: Overall network architecture of SpikeSR. The LR input is replicated along the temporal dimension and then processed through a convolution to extract shallow features. The core module of SpikeSR is SAG, which employs SABs to capture deep spiking representations. Each SAB contains three main components: (1) SCB, (2) HDA, and (3) DSA. The fusion block (FB) aggregates the spatial-temporal sequences, and pixelshuffle is used to reconstruct the SR output.

## 2 Method

The architecture of the proposed SpikeSR is illustrated in Fig. 2, which mainly consists of SAGs. Before SAGs, we utilize a $3 \times 3$ convolution to extract high-dimensional features from the LR input. These features are then processed through $m$ stacked SAGs to explore deeper representations. Each SAG includes $n$ SABs, a SCB, and a residual connection. In the SCB, leaky integrate-and-fire (LIF) neurons [37, 14] are used to convert the inputs into binary spike sequences (*i.e.*, 0 or 1). As shown in Fig. 2, the output of the LIF neuron is 1 when the membrane potential exceeds the threshold, and 0 otherwise. To optimize the membrane potential, we introduce HDA, which refines the spiking activity using an efficient temporal-channel joint attention [72]. Furthermore, the proposed DSA is employed to introduce global context for accurate SR. After the terminal SAG, an FB is utilized to convert the spike sequence features into continuous values. Finally, SpikeSR generates super-resolved output from the fused features by applying pixel-shuffle [43] and a $3 \times 3$ convolutional layer.

### 2.1 Spiking Attention Block

As evidenced in Fig. 1, regions degraded by different factors exhibit noticeable fluctuations when encoded by LIF, highlighting pronounced firing spike rates of neurons. This provides robust and latent informative spiking cues from LR images. Unlike ANNs that encode images into continuous decimal values, SNNs use discrete binary spike values for neuronal communication, and thus demonstrate undesirable information loss [24, 65], resulting in limited capacity to represent degraded LR images. To address this, the design philosophy of the SAB is focused on leveraging CNNs and attention mechanisms to regulate membrane potentials, facilitating high-quality feature representation for SR, which in turn affects the spiking activity.

As shown in Fig. 2, in particular, the output of the $n$-th SAB at the $t$-th time step is denoted as $\mathbf{X}^{t,n}$, and can be obtained by the following:

$$\mathbf{X}^{t,n} = \mathbf{X}^{t,n-1} + \text{DSA}(\text{HDA}(\bar{\mathbf{X}}_1^{t,n} + \bar{\mathbf{X}}_2^{t,n})), \tag{1}$$

where $\bar{\mathbf{X}}_1^{t,n}$ and $\bar{\mathbf{X}}_2^{t,n}$ are two feature representations obtained from parallel branches, defined by:

$$\begin{aligned}\bar{\mathbf{X}}_1^{t,n} &= \text{SCB}(\text{SCB}(\mathbf{X}^{t,n-1})), \\ \bar{\mathbf{X}}_2^{t,n} &= \text{tdBN}(\text{Conv}(\mathbf{X}^{t,n-1})),\end{aligned} \tag{2}$$

where $\text{Conv}$ represents a $3 \times 3$ convolution layer, and $\text{tdBN}$ means the threshold-dependent batch normalization.

Different from previous works that focus solely on separate temporal and channel modulation [61, 65, 45], SAB adheres to temporal-channel joint attention [72] to realize joint adjustment of the spike response in HDA, effectively achieving interdependencies between the temporal and channel scopes. More details of HDA can be found in the Appendix.

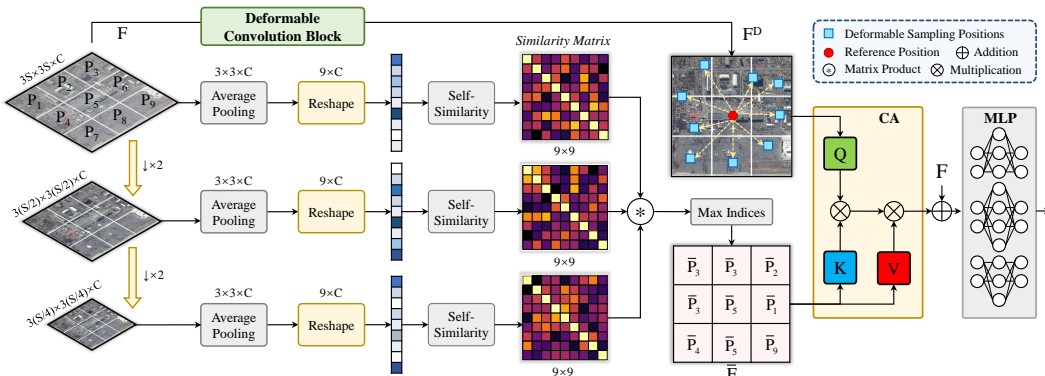

Figure 3: The illustration of our DSA. Note that we set the diagonal elements of the similarity matrix to zero before selecting the indices of the highest scores. The deformable convolution operates at the patch level, alleviating the mismatch between the most similar patches.

## 2.2 Deformable Similarity Attention

Non-local self-similarity has been recognized as an effective prior for SR tasks [47]. However, existing non-local attention mechanisms are computationally expensive due to exhaustive non-local operations, which impedes their efficiency in large-scale RSIs. In contrast, the proposed DSA efficiently grasps complex self-similar patterns in RSIs at the patch level to infer intricate spatial weights. Then, we utilize the cross-attention (CA) paradigm to enhance long-range communication, facilitating the fusion of useful context.

The details of DSA are shown in Fig. 3. Considering that the object scale exhibits explicit diversity in RSIs, the input feature F is downsampled using bilinear interpolation, forming a multi-scale feature pyramid. For clarity, we demonstrate this process by dividing the initial features into 9 patches. Following the design in [33], the final DSA exploits a cascaded patch division strategy. Specifically, each patch is first average-pooled to capture its spatial characteristics, then reshaped and subjected to self-similarity computation, yielding a similarity matrix. The final self-similarity scores are fused via matrix multiplication to enhance the multi-scale representation. The best-matching patch $\bar{\mathbf{P}}_i$ with $\mathbf{P}_i$ can be obtained by:

$$\bar{\mathbf{P}}_i = \underset{\mathbf{P}_j}{\operatorname{argmax}} \, E(\mathbf{P}_i)^{\mathrm{T}} E(\mathbf{P}_j), \;\; j \neq i, \tag{3}$$

where $\bar{\mathbf{P}}_i$ is the patch in $\bar{\mathbf{F}}$, and $E$ means the operation of average pooling and feature reshaping. We adopt the Gumbel-Softmax [52] to achieve the non-differentiable $\operatorname{argmax}$ function.

Although matched patches contain highly relevant similarity, they are inevitably subject to mismatches and geometric transformations. Hence, we use deformable convolution (DConv) to reduce the generation of hallucinated textures. The deformable feature $\mathbf{F}^{\mathrm{D}}$ at location $p_0$ is computed as follows:

$$\mathbf{F}^{\mathrm{D}}(p_0) = \sum_{p_m \in \mathcal{R}} \omega(p_m) \cdot \mathbf{F}(p_0 + p_m + \Delta p_m), \tag{4}$$

where $\omega(p_m)$ is the convolution weight at relative location $p_m$, $\Delta p_m$ is a 2D vector that represents the learnable offsets, $\mathcal{R}$ is a regular grid that determines the receptive field of the convolution kernel. For a $3 \times 3$ kernel, $\mathcal{R} = \{(-1, -1), (-1, 0), \cdots, (1, 1)\}$. To fuse the self-similar features $\mathbf{F}^{\mathrm{D}}$ with $\bar{\mathbf{F}}$, we embed $\mathbf{F}^{\mathrm{D}}$ to $\mathbf{Q}$, and $\bar{\mathbf{F}}$ to $\mathbf{K}$, $\mathbf{V}$ using fully connected layers, then perform aggregation by:

$$\bar{\mathbf{V}} = \operatorname{softmax}(\mathbf{Q}\mathbf{K}^{\mathrm{T}}/\sqrt{d})\mathbf{V}, \tag{5}$$

Finally, the fused features are summarized with the original features $\mathbf{F}$ and fed into the multilayer perceptron (MLP) to obtain the final output:

$$\tilde{\mathbf{F}} = \operatorname{MLP}(\mathbf{F} + \bar{\mathbf{V}}). \tag{6}$$

Table 1: Quantitative comparison of SpikeSR with SOTA methods on three remote sensing datasets. FLOPs are measured corresponding to an LR image of $160 \times 160$ pixels. Note that we set T = 1 to evaluate the model complexity of SpikeSR for fair comparison.

| Methods | #Param. | FLOPs | AID [56] | | DOTA [55] | | DIOR [29] | | Average | |
|---|---|---|---|---|---|---|---|---|---|---|
| | | | PSNR | SSIM | PSNR | SSIM | PSNR | SSIM | PSNR | SSIM |
| Bicubic | - | - | 28.86 | 0.7382 | 31.16 | 0.7947 | 28.57 | 0.7432 | 29.53 | 0.7587 |
| SRCNN [10] | 20K | 0.512G | 29.70 | 0.7741 | 32.10 | 0.8264 | 29.49 | 0.7768 | 30.43 | 0.7924 |
| VDSR [21] | 667K | 17.08G | 30.44 | 0.8004 | 33.22 | 0.8569 | 30.36 | 0.8036 | 31.34 | 0.8203 |
| EDSR [32] | 1518K | 50.77G | 30.65 | 0.8086 | 33.64 | 0.8648 | 30.63 | 0.8116 | 31.64 | 0.8283 |
| CARN [1] | 1112K | 40.39G | 30.66 | 0.8068 | 33.66 | 0.8633 | 30.64 | 0.8102 | 31.65 | 0.8268 |
| IMDN [20] | 715K | 18.18G | 30.71 | 0.8076 | 33.70 | 0.8641 | 30.73 | 0.8115 | 31.71 | 0.8277 |
| RFDN-L [34] | 681K | 16.49G | 30.69 | 0.8074 | 33.73 | 0.8642 | 30.72 | 0.8114 | 31.71 | 0.8277 |
| LatticeNet [36] | 777K | 19.39G | 30.73 | 0.8089 | 33.75 | 0.8653 | 30.75 | 0.8126 | 31.74 | 0.8289 |
| HNCT [13] | 364K | 8.48G | 30.79 | 0.8104 | 33.83 | 0.8664 | 30.80 | 0.8136 | 31.81 | 0.8301 |
| FMEN [12] | 1046K | 26.72G | 30.65 | 0.8063 | 33.66 | 0.8631 | 30.66 | 0.8104 | 31.66 | 0.8266 |
| RLFN [23] | 544K | 13.25G | 30.70 | 0.8074 | 33.69 | 0.8636 | 30.70 | 0.8110 | 31.70 | 0.8273 |
| ESRT [35] | 752K | 26.06G | 30.77 | 0.8102 | 33.75 | 0.8668 | 30.81 | 0.8142 | 31.78 | 0.8304 |
| SwinIR-light [31] | 897K | 23.56G | 30.83 | 0.8114 | 33.94 | 0.8677 | 30.85 | 0.8149 | 31.87 | 0.8313 |
| Omni-SR [51] | 2803K | 70.98G | 30.89 | **0.8142** | 33.94 | 0.8695 | 30.89 | 0.8170 | 31.91 | 0.8336 |
| NGswin [5] | 995K | 12.73G | 30.79 | 0.8107 | 33.87 | 0.8667 | 30.79 | 0.8140 | 31.82 | 0.8305 |
| SPIN [66] | 555K | 18.91G | 30.78 | 0.8098 | 33.85 | 0.8673 | 30.82 | 0.8139 | 31.82 | 0.8303 |
| HiT-SR [68] | 792K | 21.04G | 30.87 | 0.8138 | 33.93 | 0.8689 | 30.89 | 0.8167 | 31.90 | 0.8331 |
| SpikeSR-S (Ours) | 472K | 15.21G | 30.86 | 0.8126 | 33.89 | 0.8687 | 30.89 | 0.8162 | 31.88 | 0.8325 |
| SpikeSR-M (Ours) | 763K | 24.00G | 30.88 | 0.8133 | 33.92 | 0.8689 | 30.90 | 0.8163 | 31.90 | 0.8328 |
| SpikeSR (Ours) | 1042K | 33.05G | **30.91** | **0.8142** | **33.98** | **0.8700** | **30.95** | **0.8175** | **31.95** | **0.8339** |

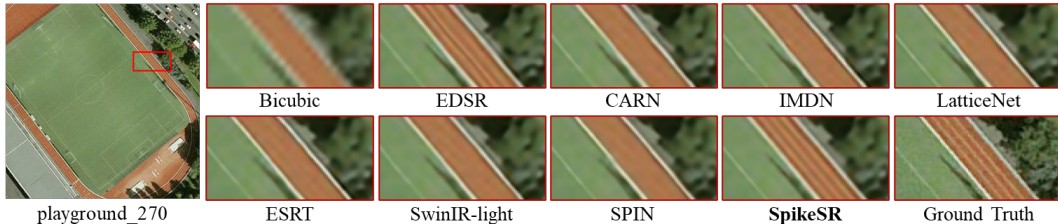

Figure 4: Qualitative comparison of SOTA efficient models for $\times 4$ SR task on AID test set.

## 2.3 Fusion Block

To transform discrete spiking sequences into continuous pixel values, a common approach is to apply mean sampling along the time dimension. However, this naive process may lead to the loss of crucial spatial details, potentially affecting the SR quality. Therefore, we introduce a fusion block that adaptively aggregates spiking sequences and mitigates information loss. Given an input spike input $\mathbf{Y}$, the computation process of FB can be formulated as:

$$\begin{aligned} \mathbf{Y}_1 &= \sigma(\text{TA}(\mathbf{Y})) \otimes \mathbf{Y}, \\ \mathbf{Y}_2 &= \sigma(\text{SA}(\mathbf{Y})) \otimes (1 - \mathbf{Y}_1), \end{aligned} \tag{7}$$

where TA and SA denote temporal and spatial attention [65], $\sigma$ means a sigmoid function and $\otimes$ denotes feature multiplication. The final output of FB is obtained by summing $\mathbf{Y}_1$ and $\mathbf{Y}_2$.

## 3 Experiments

**Datasets.** We use the AID dataset [56] as the training set, a large-scale remote sensing benchmark for scene classification, consisting of 30 different scene categories. The AID dataset includes 10,000 HR images, where we randomly select 3,000 for training and 900 for validation. The LR samples are generated by bicubic downsampling. Following TTST [58], we also evaluate our method on the DOTA [55] and DIOR [29] datasets, which contain 900 and 1,000 images, respectively.

**Implementation Details.** During model training, the learning rate is fixed to $10^{-4}$, and the training procedure stops after 1000 epochs with a batch size of 4. Adam optimizer is used with $\beta_1 = 0.9$

Table 2: Quantitative comparison of SpikeSR with SOTA methods on 30 scene types of AID datasets.

| Scene types | EDSR [32] | | CARN [1] | | IMDN [20] | | ESRT [35] | | SwinIR-L [31] | | SPIN [66] | | HiT-SR [68] | | SpikeSR | |
|---|---|---|---|---|---|---|---|---|---|---|---|---|---|---|---|---|
| | PSNR | SSIM | PSNR | SSIM | PSNR | SSIM | PSNR | SSIM | PSNR | SSIM | PSNR | SSIM | PSNR | SSIM | PSNR | SSIM |
| Airport | 29.93 | 0.8282 | 29.96 | 0.8264 | 30.00 | 0.8270 | 30.09 | 0.8292 | 30.14 | 0.8307 | 30.11 | 0.8295 | 30.21 | 0.8325 | **30.27** | **0.8336** |
| Bare Land | 36.94 | 0.8837 | 36.92 | 0.8829 | 36.93 | 0.8834 | 36.90 | 0.8840 | 36.99 | 0.8841 | 36.96 | 0.8843 | **37.00** | **0.8846** | 36.98 | **0.8846** |
| Baseball Field | 33.05 | 0.8765 | 33.06 | 0.8753 | 33.17 | 0.8763 | 33.21 | 0.8773 | 33.27 | 0.8782 | 33.14 | 0.8759 | 33.29 | 0.8791 | **33.32** | **0.8794** |
| Beach | 34.18 | 0.8727 | 34.27 | 0.8737 | 34.29 | 0.8739 | 34.35 | 0.8755 | 34.39 | 0.8756 | 34.36 | 0.8757 | 34.38 | 0.8762 | **34.41** | **0.8764** |
| Bridge | 32.93 | 0.8800 | 32.86 | 0.8774 | 32.93 | 0.8784 | 33.05 | 0.8803 | 33.15 | 0.8810 | 33.05 | 0.8803 | 33.14 | 0.8820 | **33.27** | **0.8827** |
| Center | 28.77 | 0.7921 | 28.71 | 0.7881 | 28.79 | 0.7892 | 28.88 | 0.7923 | 29.00 | 0.7954 | 28.88 | 0.7919 | 29.03 | 0.7974 | **29.09** | **0.7985** |
| Church | 26.30 | 0.7469 | 26.41 | 0.7449 | 26.46 | 0.7467 | 26.52 | 0.7489 | 26.59 | 0.7512 | 26.56 | 0.7507 | 26.64 | 0.7549 | **26.70** | **0.7560** |
| Commercial | 29.01 | 0.7940 | 29.11 | 0.7944 | 29.17 | 0.7958 | 29.22 | 0.7975 | 29.28 | 0.7996 | 29.27 | 0.7989 | 29.33 | 0.8021 | **29.37** | **0.8033** |
| D-Residential | 24.38 | 0.6839 | 24.56 | 0.6856 | 24.60 | 0.6864 | 24.67 | 0.6898 | 24.71 | 0.6924 | 24.63 | 0.6872 | 24.80 | **0.6998** | **24.80** | 0.6978 |
| Desert | 40.20 | 0.9268 | 40.22 | 0.9259 | 40.17 | 0.9264 | 40.06 | 0.9276 | **40.31** | 0.9275 | 40.24 | 0.9279 | 40.27 | 0.9282 | 40.25 | **0.9283** |
| Farmland | 35.00 | 0.8683 | 34.89 | 0.8656 | 34.94 | 0.8667 | 34.99 | 0.8679 | 35.02 | 0.8681 | 34.98 | 0.8678 | 35.09 | 0.8696 | **35.09** | **0.8700** |
| Forest | 29.85 | 0.7315 | 29.88 | 0.7304 | 29.90 | 0.7312 | 29.99 | 0.7365 | 29.98 | 0.7350 | 29.97 | 0.7350 | 30.05 | **0.7395** | **30.05** | 0.7380 |
| Industrial | 28.88 | 0.7931 | 28.84 | 0.7894 | 28.88 | 0.7904 | 28.98 | 0.7942 | 29.03 | 0.7959 | 28.98 | 0.7932 | 29.11 | 0.7998 | **29.16** | **0.8007** |
| Meadow | 34.63 | 0.7804 | 34.53 | 0.7769 | 34.55 | 0.7784 | 34.63 | 0.7814 | 34.66 | 0.7807 | 34.64 | 0.7813 | 34.58 | 0.7807 | **34.68** | **0.7820** |
| M-Residential | 28.34 | 0.7365 | 28.39 | 0.7347 | 28.42 | 0.7349 | 28.47 | 0.7370 | 28.56 | 0.7401 | 28.39 | 0.7334 | **28.64** | **0.7443** | 28.63 | 0.7436 |
| Mountain | 30.63 | 0.7885 | 30.70 | 0.7892 | 30.70 | 0.7895 | 30.74 | 0.7909 | 30.78 | 0.7921 | 30.75 | 0.7911 | **30.79** | **0.7930** | 30.79 | 0.7930 |
| Park | 30.54 | 0.8130 | 30.63 | 0.8136 | 30.65 | 0.8141 | 30.72 | 0.8169 | 30.76 | 0.8177 | 30.73 | 0.8162 | 30.81 | 0.8198 | **30.82** | **0.8203** |
| Parking | 27.25 | 0.8317 | 27.08 | 0.8245 | 27.23 | 0.8270 | 27.47 | 0.8352 | 27.42 | 0.8354 | 27.50 | 0.8363 | 27.70 | **0.8435** | **27.72** | 0.8424 |
| Playground | 35.37 | 0.8943 | 35.27 | 0.892 | 35.42 | 0.8929 | 35.47 | 0.8942 | 35.49 | 0.8946 | 35.45 | 0.8946 | 35.59 | 0.8968 | **35.70** | **0.8976** |
| Pond | 32.11 | 0.8542 | 32.10 | 0.8532 | 32.11 | 0.8534 | 32.17 | 0.8546 | 32.22 | 0.8553 | 32.15 | 0.8545 | 32.23 | 0.8561 | **32.25** | **0.8563** |
| Port | 28.50 | 0.8596 | 28.61 | 0.8593 | 28.67 | 0.8597 | 28.75 | 0.8620 | 28.81 | 0.8631 | 28.79 | 0.8624 | 28.85 | 0.8651 | **28.94** | **0.8658** |
| Railway Station | 28.72 | 0.7738 | 28.68 | 0.7699 | 28.77 | 0.7718 | 28.84 | 0.7744 | 28.92 | 0.7777 | 28.89 | 0.7759 | 28.97 | 0.7802 | **29.02** | **0.7816** |
| Resort | 28.52 | 0.7799 | 28.59 | 0.7791 | 28.62 | 0.7795 | 28.68 | 0.7819 | 28.74 | 0.7837 | 28.66 | 0.7801 | 28.78 | 0.7864 | **28.82** | **0.7869** |
| River | 31.55 | 0.7891 | 31.55 | 0.7882 | 31.57 | 0.7885 | 31.60 | 0.7900 | 31.64 | 0.7905 | 31.61 | 0.7904 | 31.66 | 0.7918 | **31.68** | **0.7922** |
| School | 29.36 | 0.8044 | 29.41 | 0.8033 | 29.45 | 0.8041 | 29.51 | 0.8067 | 29.59 | 0.8091 | 29.50 | 0.8048 | 29.67 | 0.8123 | **29.68** | **0.8124** |
| S-Residential | 27.71 | 0.6728 | 27.79 | 0.6723 | 27.80 | 0.6725 | 27.85 | 0.6752 | 27.88 | 0.6758 | 27.80 | 0.6728 | 27.91 | **0.6782** | **27.92** | 0.6775 |
| Square | 30.84 | 0.8200 | 30.83 | 0.8181 | 30.87 | 0.8183 | 30.97 | 0.8218 | 31.06 | 0.8236 | 30.98 | 0.821 | 31.11 | 0.8256 | **31.15** | **0.8266** |
| Stadium | 29.63 | 0.8387 | 29.51 | 0.834 | 29.62 | 0.8358 | 29.74 | 0.8388 | 29.82 | 0.8413 | 29.76 | 0.8394 | 29.80 | 0.8420 | **29.93** | **0.8439** |
| Storage Tanks | 27.44 | 0.7664 | 27.50 | 0.7649 | 27.52 | 0.7648 | 27.58 | 0.7671 | 27.63 | 0.7692 | 27.58 | 0.767 | 27.66 | 0.7720 | **27.68** | **0.7718** |
| Viaduct | 28.99 | 0.7757 | 28.92 | 0.7711 | 28.96 | 0.7722 | 29.06 | 0.7759 | 29.14 | 0.7784 | 29.05 | 0.7753 | 29.16 | 0.7811 | **29.25** | **0.7831** |
| Average | 30.65 | 0.8086 | 30.66 | 0.8068 | 30.71 | 0.8076 | 30.77 | 0.8102 | 30.83 | 0.8114 | 30.78 | 0.8098 | 30.87 | 0.8138 | **30.91** | **0.8142** |

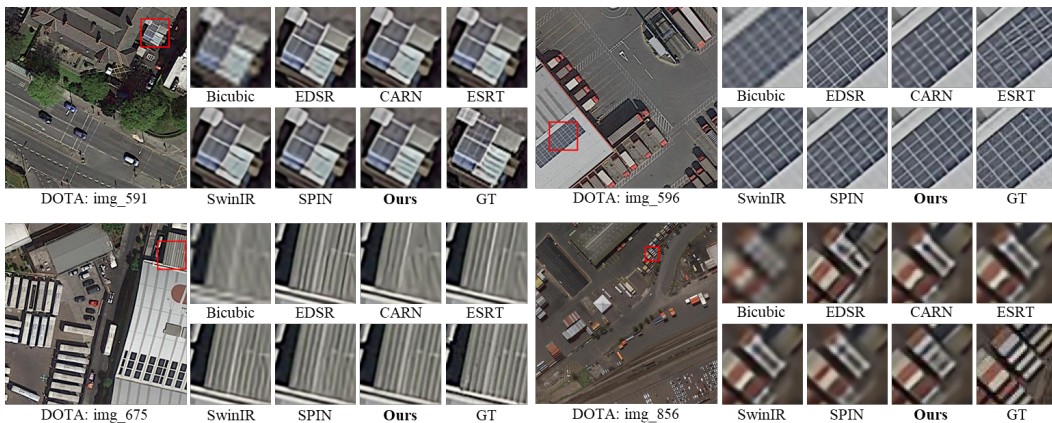

Figure 5: Qualitative comparison of SOTA efficient SR models for ×4 SR task on DOTA dataset.

and $\beta_2 = 0.999$. Data augmentation includes random rotations of 90°, 180°, 270°, and horizontal flips on $64 \times 64$ patches. The channel number, embedding dimension of cross-attention, and MLP rate of small, medium, and final SpikeSR are set to $\{40, 24, 72\}$, $\{56, 24, 72\}$, and $\{64, 32, 100\}$, respectively. The number of SAGs is 4, with 2 SABs in each SAG. Time step is set to $T = 4$.

**Metrics.** The widely used peak signal-to-noise ratio (PSNR) and structural similarity (SSIM) are used to evaluate SR performance. The results are measured on the Y channel after converting RGB to YCbCr space. For a fair comparison, all SR methods are retained on an RTX 4090 GPU from scratch using the AID dataset, adhering to their official implementation settings.

## 3.1 Comparison with Efficient Models

We compare our SpikeSR with state-of-the-art (SOTA) efficient SR methods, including CNN-based models of CARN [1], LatticeNet [36], RLFN [23], etc, and transformer-based approaches of SwinIR

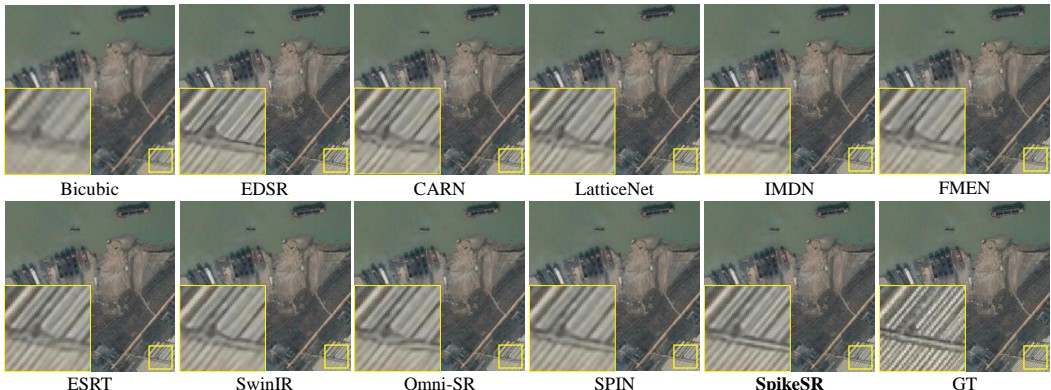

Figure 6: Qualitative comparison of SOTA efficient SR models for ×4 SR task on DIOR dataset.

[31], SPIN [66], HiT-SR [68], *etc*. We also report results for the small and medium versions of our SpikeSR, denoted as SpikeSR-S and Spike-M, respectively.

**Quantitative comparisons.** The quantitative results of various methods are reported in Table 1. We can observe that our SpikeSR achieves the best performance across three benchmarks, outperforming SOTA CNN- and transformer-based SR models. For example, on AID, DOTA, and DIOR datasets, SpikeSR improves PSNR by 0.08 dB, 0.04 dB, and 0.1 dB, respectively, compared to the impressive SwinIR. Moreover, the small version of SpikeSR requires fewer parameters (472K vs. 897K) and FLOPs (15.21G vs. 23.56G) than SwinIR, yet achieves superior average performance.

**Qualitative comparisons.** Fig. 4, Fig. 5, and Fig. 6 present visual comparisons on AID, DOTA, and DIOR datasets, respectively. As shown in Fig. 4, SpikeSR effectively restores severely damaged textures, *e.g.*, the runway line in the playground. By contrast, other SR models fail to recover such weak high-frequency details. In Fig. 5, the reconstruction of "img_591" highlights that SpikeSR produces results closest to the GT, while other methods like SPIN recovers unrealistic results. Moreover, Fig. 6 further demonstrates that SpikeSR consistently delivers superior visual quality, restoring more textural information compared to large-capacity ANN-based model like Omni-SR.

## 4   Ablation Study

We conduct ablation studies to assess the impact of key components in SpikeSR. The experimental results in Table 3 are measured on the AID-tiny dataset [58]. In particular, the Baseline model is constructed by removing the HDA and DSA and replacing them with standard temporal attention (TA), channel attention (CA), and spatial attention (SA) mechanisms. For a fair comparison, we increase the number of $m$, $n$, and channel dimensions to 8, 4, and 256, respectively, which ensures a similar number of parameters with our SpikeSR. Similarly, those settings of Variant-A are modified to 10, 5, and 128, respectively.

**Effectiveness of HDA and DSA.** Table 3 indicates how the SR performance is influenced by the HDA and DSA. Comparing the PSNR values of the Baseline and Variant-A reveals that HDA contributes a 0.12 dB improvement. This suggests that enhancing the correlation between the temporal and channel dimensions delivers better recalibration of the membrane potentials. More intuitively, we visualize the feature maps to highlight the impact of HDA, as shown in Fig. 7(b). The results illustrate that HDA effectively refines the feature representation by emphasizing salient details and suppressing irrelevant information.

By introducing the proposed DSA to the Baseline model, the PSNR results can be improved by a large margin of 0.21 dB. When employing HDA and DSA simultaneously, the resulting SpikeSR achieves an additional 0.08 dB improvement compared to Variant-B. To better demonstrate its effectiveness in capturing global self-similarity priors, we provide the LAM visualization and diffusion index in Fig. 7(a). As observed, DSA generates more pronounced LAMs and significantly increases the DI, indicating that the model activates more valuable pixels for accurate SR.

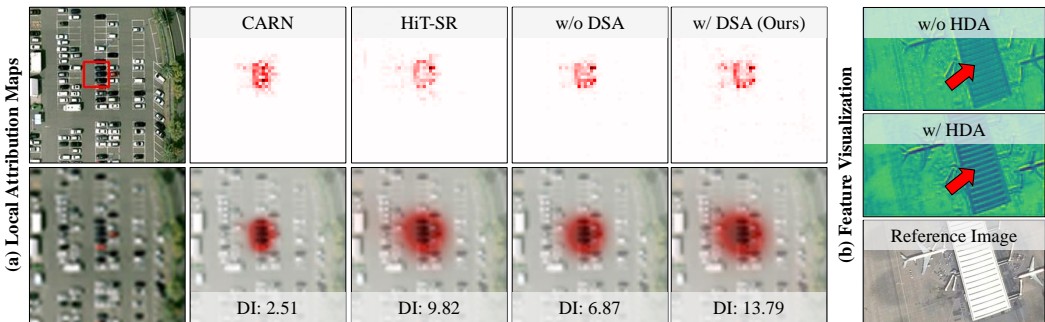

Figure 7: (a) Analysis of local attribution maps (LAMs) [16] and diffusion index (DI). The proposed DAS helps SpikeSR exploit more useful information against CARN and HiT-SR. (b) Feature visualizations. The feature obtained by HDA is sharper and preserves more details, indicating high-quality feature representations.

Table 3: Ablation on different variants of our SpikeSR.

| Methods | TA | CA | SA | HDA | DSA | #Param. | PSNR (dB) |
|---|---|---|---|---|---|---|---|
| Baseline | ✓ | ✓ | ✓ | | | 1120K | 27.80 |
| Variant-A | | | ✓ | ✓ | | 1062K | 27.92 |
| Variant-B | ✓ | ✓ | | | ✓ | 1009K | 28.11 |
| SpikeSR | | | | ✓ | ✓ | 1042K | **28.19** |

Table 4: Ablation on feature pyramid and deformable convolution.

| Methods | w/o Pyramid | w/o DConv | DSA (Full) |
|---|---|---|---|
| #Param. | 1042K | 918K | 1042K |
| FLOPs | 31.41G | 28.86G | 33.00G |
| PSNR (dB) | 28.14 | 28.07 | **28.19** |

Table 5: Performance and complexity analysis of SpikeSR with different numbers of blocks.

| Blocks | 2 | 3 | 4 | 5 | 6 |
|---|---|---|---|---|---|
| #Param. | 558K | 800K | 1042K | 1284K | 1526K |
| FLOPs | 17.47G | 25.26G | 33.00G | 40.84G | 48.60G |
| PSNR (dB) | 28.11 | 28.17 | **28.19** | 28.16 | **28.20** |

**Feature Pyramid and DConv.** Due to the scale diversity of objects in RSIs, we constructed a feature pyramid to grasp the self-similarity in multiple levels. As listed in Table 4, the use of feature pyramid improves the performance by 0.05 dB without introducing additional parameters. Furthermore, to demonstrate the effectiveness of deformable convolution, we remove this component, which leads to a severe performance drop by 0.12 dB. This illustrates that the self-similar patches contain massive irrelevant and misaligned contents, and direct fusion may introduce interference, thus resulting in suboptimal performance.

**Network Depth.** We evaluate the impact of the network depth by changing the number of SAGs of our SpikeSR from 2 to 6 blocks. As reported in Table 5, SpikeSR achieves the highest SR performance when $m = 3$. While increasing the number of $m$ may further improve the reconstruction, it also brings larger model size. Therefore, we set $m = 4$ in our final model, considering the trade-off between performance and computational complexity.

## 5   Conclusion and Limitation

In this paper, we investigate the application of SNNs for efficient SR of remote sensing images. Motivated by the observation that LIF neurons exhibit a higher spike rate in degraded images, we integrate SNNs with convolutions for improved feature representation. Besides, a hybrid dimension attention is employed to modulate the spike response, further refining salient information. To incorporate valuable prior knowledge for more accurate SR, we propose a deformable similarity attention module, capturing global self-similarity across multiple feature levels. Extensive experiments on various remote sensing datasets demonstrate the efficacy and effectiveness of the proposed SpikeSR model. We believe our exploration can facilitate the practical application of energy-efficient models in remote sensing area.

SpikeSR still has some limitations. First, its representational capacity remains improvable. Second, the reliance on attention mechanisms introduces additional computational overhead, limiting efficiency. We leave these issues for future exploration.

**Acknowledgment.** This work is supported in part by the National Natural Science Foundation of China (423B2104, 623B2080), and in part by the Natural Science Foundation of Heilongjiang Province of China for Excellent Youth Project (YQ2024F006), and in part by the Open Research Fund from Guangdong Laboratory of Artificial Intelligence and Digital Economy (SZ) (GML-KF-24-09).

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
