# OpenReview forum: "Spiking Meets Attention: Efficient Remote Sensing Image Super-Resolution with Attention Spiking Neural Networks"
_NeurIPS.cc/2025/Conference — NeurIPS 2025 poster_

### Official Review · Reviewer_WHSi · 2025-07-02

**Clarity:** 4
**Significance:** 4
**Originality:** 3
**Rating:** 5
**Confidence:** 5

**Summary:**

This work presents an innovative framework (SpikeSR) for remote sensing image super-resolution, where spiking neural networks are combined with attention mechanisms, providing a well-motivated SNN-based paradigm for improved efficiency and SR quality. To start with, the authors reveal the SNN's inherent properties for handling image degradation and highlight two challenges in adapting SNN to SR -- pixel-wise information loss and suboptimal membrane potential dynamics. To mitigate these issues, a Spiking Attention Block (SAB) is proposed by dynamically regulating membrane potentials through inferred attention weights, thus enhancing feature representation. Based on SAB, the proposed SpikeSR demonstrates promising performance compared against SOTA methods, as validated by comprehensive ablation and experiments over multiple datasets.

**Questions:**

Please check the Weaknesses above on the usage of deformable convolution, ablation on the time step T, more clarification on Fig.1a and details on AID-tiny.

It is also recommended to correct some typos (though few in the work), for example, typos following the values 90, 180, and 270.
Possibly a title for Sec.1 introduction is missed.

**Ethical Concerns:**

["NO or VERY MINOR ethics concerns only"]

**Final Justification:**

Thanks to the author of the rebuttal, and the extended materials help address my concerns. I am willing to keep my original rating of accept.

**Limitations:**

Yes.

**Paper Formatting Concerns:**

N/A.

**Quality:**

4

**Strengths And Weaknesses:**

Strengths:
1. The authors make the first attempt of SNN-based RSI SR, and correctly highlight that the limited representation capacity of SNNs has hindered their application in SR tasks, which makes it a valuable and highly original contribution to the field.
2. The observation of active learning states in spiking signals and their connection to the attention modulation adds valuable understanding to SNN capabilities, providing a new perspective on developing efficient models in large-scale Earth observation scenarios.
3. Achieving performance comparable to or even surpassing that of ANNs using SNNs is non-trivial. SpikeSR outperforms advanced ANN-based approaches and has been thoroughly validated on multiple large-scale remote sensing benchmarks, demonstrating a significant technical superiority.
4. The proposed SpikeSR is concise yet effective. The manuscript is well-organized and clearly written, making it easy to follow.

Weaknesses:
1. It is recommended to clarify whether the deformable convolution is applied at the patch level or the full-image level.
2. The number of time steps T is a crucial parameter for SNNs. Please consider providing more ablation on different selection of T (e.g., 1, 2, 4, 8) for analyzing its impact.
3. Does the pixel-level intensity smoothing observed in Fig. 1a imply a reduced ability of the ANN-based model to capture fine-grained details. This is intuitively important for understanding the motivation of the proposed SpikeSR and should be further discussed or explicitly pointed out.
4. In the ablation section, the details for the AID-tiny used by the authors are unclear. It is recommended to release this dataset publicly to ensure experimental reproducibility.

---

> ### Author Rebuttal · Authors · 2025-07-28
>
> ## **Response to Reviewer WHSi**
>
> Dear Reviewer WHSi:
>
> We sincerely thank the reviewer for the encouraging and insightful feedback. We are pleased that you consider our first attempt as **a valuable and highly original contribution** to the field, and our observation of active learning states **provides a new perspective** in large-scale Earth observation scenarios. We are especially encouraged by your recognition that our performance gain against ANN-based models is **non-trivial**, demonstrating a **significant technical superiority**. We hope our responses below can address your concerns.
>
> ### Weaknesses
>
> **W1: Clarify the application level of deformable convolution.**
>
> A1: To save computational cost, deformable convolution is applied at the image level. This enables our model to flexibly mitigate geometric transformations across entire images while maintaining efficiency. We will clarify this detail in the revised manuscript!
>
> **W2: Please provide more ablation on different selections of T.**
>
> A2: To assess its impact, we conducted additional ablation studies with T=1,2,4,8. The results are presented below. It can be observed that as T increases, more spatiotemporal information is incorporated, leading to progressively improved performance. However, this improvement saturates when T=4. Considering that model complexity also grows with increasing T, we choose T=4 as the final setting to strike a favorable balance between performance and efficiency. This ablation will be included in the Appendix!
>
> | Time Steps| 1| 2| 4 | 8 |
> | :---: | :---: | :---: | :---: | :---: |
> | FLOPs (G) | 32.45  | 32.78 | 33.05 | 33.31 |
> |PSNR (dB) | 27.98 | 28.10 | **28.19** | 28.17 |
>
> **W3: More clarification of pixel-level intensity observed in Fig. 1a should be discussed.**
>
> A3: Yes, the smoothed pixel-level intensity indeed reflects the tendency of ANN-based architecture (e.g., convolution kernel) to over-smooth local textures, which may hinder their ability to capture fine-grained spatial details. This phenomenon highlights the importance of temporal dynamics and sparse activation in enhancing spatial sensitivity. To clarify this, we will revise the caption of Fig. 1 to explicitly point out that the smoothing effect serves as an intuitive motivation for adopting SNN architectures in SR tasks.
>
> **W4: The details of AID-tiny are unclear.**
>
> A4: Thanks for your suggestion. The AID-tiny is a lightweight dataset derived from the AID dataset, where we randomly select one image from each of the 30 categories, resulting in a total of 30 samples. These samples are non-overlapping with the training and testing sets. To ensure reproducibility, we will release the AID-tiny dataset publicly on our GitHub repository.
>
> ### Questions
>
> **Q1: Some typos should be corrected.**
>
> A1: Thank you for pointing this out. We have carefully reviewed the manuscript and corrected the noted typos, including those following the values 90, 180, and 270. In addition, we have added the missing section title for Section 1. These revisions will be reflected in the updated manuscript!

---

> > ### Comment · Reviewer_WHSi · 2025-08-05
> > **A satisfactory responses with all concerns addressed.**
> >
> > Thanks for the detailed responses, and all my concerns are addressed.
> >
> > I also read through the comments by other reviewers, and incorporating some of the responses to the manuscript may help clarify the novelty and strength of the proposed method. For example,
> >
> > 1. The response to W1 from DFR5 gives a better understanding of energy consumption, which validates the potential for remote sensing images for its greater scale and expansion in usage for diverse earth observation applications.
> >
> > 2. The two new SOTA methods provide a better view on the development of SR and a well benchmark for performance comparison.
> >
> > 3. The ablation on T help on demonstrating the performance robustness over parameter selections.
> >
> > Based on the reasons above, I would keep my rating of `accept`.

---

> > > ### Author Response · Authors · 2025-08-05
> > >
> > > Thank you for your positive comments and support!

---

> ### Author Response · Authors · 2025-08-05
>
> Dear reviewer,
>
> Thank you for the comments on our paper.
>
> We have submitted the responses to your comments. Please let us know if you have additional questions so that we can address them during the discussion period.
>
> Thank you!
>
> Best,
>
> Authors

---

### Official Review · Reviewer_Ecnp · 2025-07-02

**Clarity:** 2
**Significance:** 3
**Originality:** 3
**Rating:** 3
**Confidence:** 4

**Summary:**

This paper introduces SpikeSR, a novel framework for remote sensing image super-resolution based on Spiking Neural Networks (SNNs). SpikeSR leverages the energy efficiency and sparsity of Leaky Integrate-and-Fire (LIF) neurons to reduce computational overhead, while incorporating attention-based mechanisms to enhance feature representation in degraded low-resolution images. The proposed method includes several key components: a Hybrid Dimension Attention (HDA) module, which recalibrates spiking responses across temporal and channel dimensions to facilitate joint feature correlation learning; a Deformable Similarity Attention (DSA) module, which captures multi-scale spatial self-similarity via deformable attention to align and integrate relevant textures; and a Fusion Block that adaptively merges spike sequences guided by both temporal and spatial attention to mitigate information loss. Experiments on the AID, DOTA, and DIOR datasets demonstrate that SpikeSR achieves superior performance compared to existing CNN- and Transformer-based SR methods, while maintaining high computational efficiency.

**Questions:**

Clarification on the HDA module: The Hybrid Dimension Attention (HDA) is a core component of your model, yet its description in the main text is quite limited. While you mention that further details are provided in the appendix, the current version of the submission does not appear to include any appendix. For reproducibility and a clearer understanding of the model’s contribution, please provide a detailed explanation of the HDA module in the main paper or ensure the appendix is available and sufficiently descriptive. Clarifying this could significantly strengthen the clarity and transparency of the proposed method.

The neuron voltage visualization in Figure 1(a) is an interesting perspective. However, the connection between spiking responses and actual SR reconstruction quality remains somewhat speculative. Could the authors clarify whether regions with high voltage activity (as shown)  correspond to better preserved or recovered high-frequency content in the final SR outputs?

A key motivation for using SNNs lies in their ability to capture temporal dynamics. However, the paper lacks any systematic analysis of how these dynamics—such as the number of simulation steps or LIF parameters—affect performance. Ablation studies or controlled experiments along this axis would significantly strengthen the argument that the model benefits from intrinsic spiking properties.

While the architecture is thoughtfully designed and the integration of components is well-executed, it largely builds upon existing ideas—spiking neurons combined with attention mechanisms such as temporal-channel recalibration and deformable similarity. The paper would benefit from a more detailed explanation of what unique representational benefits or inductive biases this combination offers. At present, the novelty appears to lie more in the architectural composition than in a fundamentally new conceptual insight.

**Ethical Concerns:**

["NO or VERY MINOR ethics concerns only"]

**Final Justification:**

Thank you for the authors’ rebuttal. However, after considering all the other reviews and rebuttals to them, I have decided to maintain my original score.

**Limitations:**

Yes

**Quality:**

2

**Strengths And Weaknesses:**

1. Quality
 While the paper presents a technically sound framework and demonstrates solid empirical performance, its contribution would be further strengthened by a deeper exploration of the unique characteristics of Spiking Neural Networks (SNNs). The current architecture—based on a standard LIF-based spiking backbone with Transformer-inspired attention modules—largely follows a modular design pattern, and it remains somewhat unclear to what extent it fully capitalizes on the temporal and sparse properties of SNNs. The proposed HDA and DSA modules are effective, yet their connection to the underlying spiking dynamics appears limited, and similar designs could potentially be adapted to non-spiking models as well. Additionally, although the motivation highlights the temporal modeling capabilities of SNNs, the paper provides limited analysis regarding how specific temporal factors—such as spike timing, membrane integration behavior, or variations in time steps—impact reconstruction performance. A more thorough investigation along these lines could clarify the specific advantages offered by spike-based representations in the SR context.
2. Clarity
The manuscript is generally clear and well-organized: the motivation is well-articulated and the problem formulation is clear. The architecture diagrams (especially Fig. 3) are informative and visually aid understanding. However, the description of the key component, the Hybrid Dimension Attention (HDA) module, is relatively brief in the main text. Although the authors mention that more details are provided in the appendix, including a more comprehensive explanation of its structure and working mechanism in the main body would significantly enhance readability and overall clarity.
3. Originality
While the application of Spiking Neural Networks (SNNs) to image super-resolution is relatively novel, the architectural innovation of this work appears to be incremental. The proposed framework follows a modular “SNN backbone + attention mechanism” design, with its core components—such as deformable attention and dimension-wise recalibration—drawing from established techniques in CNN and Transformer literature. As a result, the contribution seems to focus more on the integration and adaptation of known methods .Further investigation into how spiking dynamics interact with SR-specific challenges—such as texture restoration or temporal information flow—could enhance the originality and domain relevance of the work.
4. Significance
While the method demonstrates solid empirical performance with low computational overhead, its broader impact is somewhat constrained by the limited exploration of the underlying mechanisms that make spiking neural networks effective for super-resolution. That said, the direction itself—adapting spike-based computation to dense prediction tasks such as SR—is timely and meaningful. In a field where energy efficiency and model compactness are increasingly important, the attempt to leverage the temporal dynamics and sparsity of SNNs is conceptually promising. Therefore, although the current implementation may lack theoretical depth, the research direction it represents is worth exploring, as it has the potential to develop into a more generalizable and insightful solution and drive further progress in the field of efficient vision models.

---

> ### Author Rebuttal · Authors · 2025-07-28
>
> ## **Response to Reviewer Ecnp**
>
> Dear Reviewer Ecnp:
>
> Thank you for your encouraging comments and insightful questions. We are glad that you found our method **technically sound**, and recognized the application of SNN to SR as **relatively novel** and **conceptually promising**. We also appreciate your recognition of the **well-articulated motivation** behind our work, and your acknowledgment that our research direction is **timely, meaningful, and worth exploring**, which highlights the originality and potential impact of our contribution. We hope that the responses below will address your concerns, and we would be truly grateful if you could consider updating the score!
>
> ### Weaknesses and Questions
>
> **W1 & Q3: The paper would benefit from more ablation of time steps and connection to the underlying spiking dynamics.**
>
> A1: **To better understand the influence of time steps:** we conducted additional ablation experiments by setting different numbers of simulation steps T={1, 2, 4, 8}. As shown in the table below, increasing T allows our model to integrate more spatiotemporal cues, which leads to increased performance improvements. However, this gain saturates around T=4, suggesting that further increasing T may introduce redundant or noisy information. Moreover, a larger T also leads to additional computational overhead. Therefore, we selected T=4 as a practical trade-off between efficiency and reconstruction quality. These ablations will be included in the Appendix to strengthen the argument for the effectiveness of spike-based representations in the SR context.
>
> | Time Steps| 1| 2| 4 | 8 |
> | :---: | :---: | :---: | :---: | :---: |
> | FLOPs (G) | 32.45  | 32.78 | 33.05 | 33.31 |
> |PSNR (dB) | 27.98 | 28.10 | **28.19** | 28.17 |
>
> **To further clarify the connection between HDA and DSA modules and the spiking temporal dynamics:** we conducted an in-depth visualization of feature and spiking representations. First, HDA effectively filters out redundant spike activations along both temporal and channel dimensions, thereby enhancing the quality of feature representations, as illustrated in Fig. 7(b). Furthermore, we additionally visualized spiking maps at different time steps and observed that, with the guidance of DSA, the temporal signals exhibit notable variation across different time steps. This indicates that DSA significantly enriches the temporal dynamics and mitigates the homogenization issue commonly observed in SNNs over the temporal dimension. These visualization results and analyses will be included in the supplementary materials!
>
> **W2 & Q1: More explanation of HDA could significantly strengthen the clarity.**
>
> A2: In the revised main body of our paper, we will describe the working mechanism of the HDA module and briefly introduce its structure. Specifically, HDA first applies a squeeze operation to compress membrane potentials into a compact matrix. It then performs 1D convolutions independently along the temporal and channel dimensions to extract attention weights that reflect the dynamic importance of features across both axes. To model their interdependence, we perform element-wise multiplication to fuse the two attention weights into a unified temporal-channel attention map, which is ultimately used to modulate the membrane potential representations. Instead of operating directly on raw spike trains, HDA enhances feature representation by adaptively adjusting membrane potentials, enabling it to capture rich spatiotemporal information while preserving the discrete nature of SNNs.
>
> Moreover, we will release the source code to ensure technical completeness and reproducibility of HDA. Also, we will ensure the appendix is available and sufficiently descriptive.
>
> **W3: More investigation of spiking dynamics.**
>
> A3: We have added more visual analyses to illustrate how spike-based temporal information flow facilitates texture restoration and detail propagation. Specifically, we present visualizations of spiking maps across different time steps, which reveal that the proposed DSA module enhances the variation of spiking dynamics at each temporal stage by introducing self-similarity priors.
>
> Moreover, features across different time steps exhibit distinct levels of detail saliency, which aligns with the biological inspiration of SNNs—where the human brain selectively attends to different information at different moments. After removing the HDA module,  we observe a noticeable loss of high-frequency textures and more homogeneous patterns across time steps. These visualizations further validate the effectiveness of our model in addressing the key challenges of super-resolution.
>
> **Q2: The connection between spiking responses and actual SR reconstruction quality in Fig.1 should be clarified.**
>
> We sincerely appreciate your recognition of Figure 1 as **an interesting perspective**.
>
> As clarified, the high-voltage regions do not directly correspond to high-frequency areas in the final SR outputs. Rather, the visualization in Fig. 1 is intended to illustrate our motivation, not to serve as direct empirical evidence. Specifically, it demonstrates that spiking neurons can sustain an active state even when high-frequency details are reduced. In contrast, intuitively, convolutional networks tend to exhibit reduced activation in response to degraded high-frequency inputs, which may limit their representational capacity. We will revise the caption of Fig. 1 to better clarify this point and avoid potential misinterpretation.

---

> ### Author Response · Authors · 2025-08-05
>
> Dear reviewer,
>
> Thank you for the comments on our paper.
>
> We have submitted the responses to your comments. Please let us know if you have additional questions so that we can address them during the discussion period.
>
> Thank you!
>
> Best,
>
> Authors

---

> ### Author Response · Authors · 2025-08-09
> **Any Additional Concerns?**
>
> Dear Reviewer Ecnp:
>
> Many thanks for your time!
>
> We noticed there has been no response to our feedback, and would like to kindly confirm whether our updates have addressed your concerns.
>
> As the discussion phase is approaching its end, please be assured that we are ready to clarify any remaining questions you might have!
>
> Best regards,
>
> All authors

---

### Official Review · Reviewer_DFR5 · 2025-07-03

**Clarity:** 3
**Significance:** 2
**Originality:** 2
**Rating:** 3
**Confidence:** 5

**Summary:**

This paper presents SpikeSR, a novel framework based on Spiking Neural Networks (SNNs) for remote sensing image super-resolution. The authors propose a Spiking Attention Block (SAB) that effectively optimizes membrane potentials through inferred attention weights, thereby regulating spiking activity for better feature representation. The key contributions of this work include bridging the modulation between temporal and channel dimensions and leveraging global self-similarity patterns in large-scale remote sensing scenarios to infer spatial attention weights. The proposed method achieves state-of-the-art performance across various remote sensing benchmarks while maintaining high computational efficiency.

**Questions:**

Could the authors provide a more detailed theoretical analysis of the proposed SAB, particularly regarding how the inferred attention weights influence the optimization of membrane potentials and the subsequent improvement in feature representation?

**Ethical Concerns:**

["NO or VERY MINOR ethics concerns only"]

**Final Justification:**

I appreciate the authors' detailed response. However, I must note that it did not adequately address my concerns. Specifically, the current manuscript appears to simply apply SNNs to remote sensing super-resolution without providing meaningful insights specific to this task. In my view, this work does not yet meet the threshold for NeurIPS. I am willing to raise the score by one point to a borderline rejection, but my overall assessment remains negative.

**Limitations:**

Yes

**Quality:**

2

**Strengths And Weaknesses:**

Pros:

1. Remote sensing image super-resolution has practical applications.

2. The paper was well written and organized.

3. Validation was performed on various datasets and ablation experiments were performed.

Cons:

1. Lack of clear motivation. From the current manuscript, it is not clear the advantages of applying SNN to remote sensing images of SR. What are its benefits over lightweight SR networks?

2. Does not show significant novelty. It appears that this work simply combines SNN with an attention mechanism where some simple improvements are made to the attention mechanism.

3. Performance is not good enough. This work suffers from a performance disadvantage even when compared to HiT-SR for natural images. In fact, there are many cutting-edge SR methods that are not included in the comparison.

4. The biggest advantage of SNN is low energy consumption. There is a complete lack of energy consumption related comparisons in the table.

5. Regarding inference efficiency, memory overhead related comparisons are missing.

---

> ### Author Rebuttal · Authors · 2025-07-28
>
> ## **Response to Reviewer DFR5**
>
> Dear Reviewer DFR5:
>
> Thank you for your insightful comments. We are pleased that you think our SpikeSR is **a novel framework**, recognize its **practical applications**, and consider our manuscript **well-written** and **well-organized**. We appreciate your recognition of its achievement of **state-of-the-art performance** across multiple remote sensing benchmarks, while maintaining **high computational efficiency**. We hope our responses below address your concerns, and we kindly ask you to consider updating the score.
>
> ### Weakness
>
> **W1: What are the benefits over lightweight SR networks?**
>
> A1: Thank you for your valuable feedback. Compared to lightweight SR networks, SNNs offer unique advantages due to their biological plausibility and sparse spatiotemporal dynamics. Thus:
>
> 1) These characteristics enable SNNs to achieve **significantly lower energy consumption** (as listed in the table below), which motivates us to apply SNN to reconstruct large-scale and high-resolution remote sensing images.
>
> 2) We observe spiking neurons maintain an active learning state across low-resolution images, even in severely damaged textures, which motivates us to leverage the inherent properties of SNNs to handle image degradation for efﬁcient yet high-quality SR.
>
> 3) With the growing interest and demand for in-orbit processing of remote sensing data,  our method enjoys highly energy-efficient benefits and favorable performance, making it a promising solution for practical scenarios.
>
> | Methods | EDSR | CARN | SwinIR-light | OmniSR | NGswin | SPIN | HiT-SR | SpikeSR |
> | :---: | :---: | :---: | :---: | :---: | :---: | :---: | :---: | :---: |
> | Energy (pJ)  | 162.46 | 129.25 | 70.39 | 227.14 | 40.74 | 60.51 | 67.33 | **4.74** |
> | PSNR (dB)  | 31.64 | 31.65 | 31.87 | 31.91 | 31.82 | 31.82 | 31.90 | **31.95** |
>
> We will polish the motivation in the revised manuscript to explicitly highlight these benefits!
>
> **W2: Concerns about the novelty**
>
> A2: Thank you for your thoughtful comment. We appreciate the opportunity to clarify the novelty and significance of our work：
>
> **1. The first attempt to bridge research gap for SNN-based SR**
>
> While our framework indeed integrates these components, we emphasize that it is **the first attempt** to successfully adapt and customize attention mechanisms specifically to the SNNs for SR tasks.
>
> Although SNNs are known for their energy efficiency, they have rarely been explored in remote sensing SR tasks due to their limited representation power compared to ANN-based models. This is primarily due to the inherent difficulty of adapting *continuous, dense* attention operations to the *discrete, sparse* spiking signals and their temporal dynamics, which pose unique architectural and training challenges.
>
> Our work is the first to mitigate this gap, achieving ANN-comparable or superior performance while maintaining significantly low energy consumption.  This novelty is recognized by Reviewer WHSi, who states our approach *makes a valuable and highly original contribution to the field* and *provides a new perspective on developing efficient models*, and Reviewer bD3C, who highlights *for the first time* and *filling the gap of SNN in the field of remote sensing SR*, further affirming the impact and novelty of our contribution.
>
> **2. Concise is not simple: Our method is concise yet highly effective**
>
> Although our method is concise, it is far from simplistic. Our design achieves significant performance improvements with modest architectural modifications, demonstrating that intuitive and efficient solutions can be highly effective.
> As reported in the table below, the performance improvement demonstrates that simple yet spike-compatible attention mechanisms can improve the limited capacity of SNN by a large margin, making it competitive to ANN-based SR methods. We believe this aligns with the broader research ethos that: *a well-justified, efficient improvement is often more impactful than a complex, over-parameterized solution.* Reviewer WHSi also acknowledges that “*achieving performance comparable to or even surpassing that of ANNs using SNNs is **non-trivial**.*”
>
> | Methods | EDSR | CARN | SwinIR-light | OmniSR | NGswin | SPIN | HiT-SR | SpikeSR |
> | :---: | :---: | :---: | :---: | :---: | :---: | :---: | :---: | :---: |
> | Venus  | CVPRW'17| ECCV'28 | ICCV'21| CVPR'23| CVPR'23| ICCV'23| ECCV'24| - |
> | PSNR (dB)  | 31.64| 31.65| 31.87 | 31.91 | 31.82 | 31.82 | 31.90 | **31.95** |
>
> **3. Customized technical novelty for attention-based SNNs**
>
> While the design of combining SNN backbones with attention might seem straightforward, our contribution lies in the careful adaptation and customization of attention modules (HDA and DSA) to the unique spike-based characteristics. In particular:
>
> 1) Our HDA and DSA operate on membrane potentials rather than dense activations, aligning with the discrete and temporal nature of spiking dynamics;
>
> 2) By exploring joint temporal and channel-wise dependencies within the sparse spiking sequences, we achieve a performance improvement of +0.07dB, compared to naive temporal and channel attention mechanisms.
>
> 3) Developing a pyramid structure to exploit rich multi-scale self-similarity priors. This design significantly enhances the performance against conventional spatial attention, achieving an improvement of +0.27 dB.
>
> As reviewer WHSi notes: *our technical superiority is significant*.
>
> We hope that the above clarifications can address your concerns, and we sincerely look forward to your updated score.
>
> **W3: Concerns about the performance**
>
> A3: Thank you for raising this valuable concern. We would like to clarify that our SpikeSR does not exhibit a performance disadvantage compared to HiT-SR. While our model has significantly fewer parameters (472K vs. 792K), it achieves a slightly lower performance (31.88 vs. 31.90), which is reasonable given the smaller model size. When comparing models with similar parameters (763K vs. 792K), our approach delivers comparable results. Furthermore, upon scaling up the model, SpikeSR outperforms HiT-SR (31.95 vs. 31.90), demonstrating the superiority and scalability of our framework.
>
> To further highlight the favorable performance of our SpikeSR, we have conducted additional comparisons with two SOTA efficient SR methods, **SeemoRe (ICML 2024)** and **CATANet (CVPR 2025)**. As shown below, SpikeSR remains highly competitive against these cutting-edge approaches. We will add these results to our manuscript!
>
> | Methods | SeemoRe | CATANet | SpikeSR |
> | :---: | :---: | :---: | :---: |
> | AID | 30.83/0.8115 |  30.88/0.8127 | **30.91/0.8142** |
> | DOTA | 33.94/0.8671 | **33.99**/0.8688 | 30.98/**0.8700** |
> | DIOR | 30.84/0.8141 | 30.92/0.8162 | **30.95/0.8175** |
> | Average | 31.87/0.8309 | 31.93/0.8326 | **31.95/0.8339** |
>
> **W4 & W5: Missing energy consumption and memory overhead**
>
> A4 & A5: We have additionally evaluated these metrics to address this concern. As shown in the table below, our SpikeSR demonstrates **significantly lower energy consumption** compared to ANN-based methods, highlighting the inherent energy efficiency advantage of SNNs. We will include a comprehensive discussion on model efficiency in the revised manuscript!
>
> | Methods | EDSR | CARN | SwinIR-light | OmniSR | NGswin | SPIN | HiT-SR | SpikeSR-S | SpikeSR-M | SpikeSR |
> | :---: | :---: | :---: | :---: | :---: | :---: | :---: | :---: | :---: | :---: | :---: |
> | Energy (pJ)  | 162.46 | 129.25 | 70.39 | 227.14 | 40.74 | 60.51 | 67.33 | **2.96** | 4.15 | 4.74 |
> | Memory (MB)  | 202.00 | 6.00 | 126.00 | 146.00 | 10.00 | **4.00** | **4.00** | 6.00 | 10.00 | 14.00 |
>
> **Q1: Provide a more detailed theoretical analysis of the proposed SAB**
>
> A1: Thank you for raising this insightful question. We provide a more in-depth theoretical analysis of the proposed SAB. Specifically, our SAB is composed of two key modules: Hybrid Dimension Attention (HDA), which operates in *temporal-channel joint dimensions*, and Deformable Similarity Attention (DSA), which adjusts membrane potentials across *spatial dimensions*. Together, they serve to modulate the dynamics of membrane potential accumulation in a spike-compatible manner, thus improving information flow and representation quality.
>
> Mathematically, in standard Leaky Integrate-and-Fire (LIF) neuron models, the membrane potential $u_t$ of neuron $i$ at time
> $t$ evolves as: $u_t^i=\tau u_{t-1}^i+x_t^i-V_{\mathrm{th}}s_{t-1}^i$, where $\tau$ is the decay constant, $x_t^i$ is the input current, $s_{t-1}^i$ is the spike signal from the previous time step, and $V_{\mathrm{th}}$ is the firing threshold. Without attention guidance, membrane potential integration is uniform, lacking selective representation in time or importance across channels and positions, which significantly limits the representation power of SNN.
>
> To address this, we introduce attention-guided modulation over the membrane potential update: $u_t=\tau u_{t-1}+\alpha_t\odot x_t+\beta_t$. Here, $\alpha_t\in\mathbb{R}^C$ is a joint temporal-channel attention map from HDA, $\beta_t\in\mathbb{R}^{H\times W}$ is a spatial modulation map from DSA. In this form, HDA re-weights input activations across channels and time, focusing accumulation on informative temporal-channel combinations. DSA refines the spatial saliency, emphasizing regions that contribute more meaningfully to super-resolution. The refined update thus becomes: $u_t^{i,j}=\tau u_{t-1}^{i,j}+\alpha_t^c\cdot x_t^{i,j,c}+\beta_t^{i,j}$.
>
> In the backward pass, the pseudo-gradient of loss L w.r.t. the membrane potential is: $\frac{\partial L}{\partial u_t}\approx\frac{\partial L}{\partial s_t}\cdot\sigma^{\prime}(u_t-V_\mathrm{th})$. Due to attention-induced focus in $u_t$, more informative gradients are passed to salient positions and time steps, yielding stronger feature representation power.
>
> We will include this theoretical analysis in the Appendix!

---

> ### Author Response · Authors · 2025-08-05
>
> Dear reviewer,
>
> Thank you for the comments on our paper.
>
> We have submitted the responses to your comments. Please let us know if you have additional questions so that we can address them during the discussion period.
>
> Thank you!
>
> Best,
>
> Authors

---

> > ### Comment · Reviewer_DFR5 · 2025-08-05
> >
> > Thank you for your response.
> >
> > 1. I still don't see the connection between this research and remote sensing. What are the differences between remote sensing scenes and natural image SR, and how are these issues addressed? Why are the comparison methods all natural image SR methods?
> >
> > 2. There have been many studies on SNN-based Transformers [1,2,3], and the proposed method does not show significant differences compared to these works.
> >
> > 3. Why not compare with strong methods like MambaRV2?
> >
> > [1] Spike-driven Transformer. NeurIPS 2024.
> > [2] Spike-driven Transformer V2: Meta Spiking Neural Network Architecture Inspiring the Design of Next-generation Neuromorphic Chips. ICLR 2024.
> > [3] Scaling spike-driven transformer with efficient spike firing approximation training. IEEE TPAMI 2025.

---

> > > ### Author Response · Authors · 2025-08-07
> > > **Additional Comparison with ESTNet and MambaIRv2**
> > >
> > > Dear Reviewer DFR5:
> > >
> > > Thank you once again for your valuable and constructive feedback.
> > >
> > > We have conducted additional experiments with **ESTNet (TIP 2024)** and **MambaIRv2 (CVPR 2025)**. The results are as follows:
> > >
> > > | Methods | ESTNet | MambaIRV2 | SpikeSR |
> > > | :---: | :---: | :---: | :---: |
> > > |  AID | 30.81/0.8110 | **30.92**/0.8135 | 30.91/**0.8142**|
> > > | DOTA  | 33.88/0.8673  | **34.04**/0.8695  | 33.98/**0.8700** |
> > > |  DIOR | 30.87/0.8162 | **30.96**/0.8169 | 30.95/ **0.8175**|
> > > |  Average | 31.85/0.8315 | **31.97**/0.8333 | 31.95/**0.8339**|
> > >
> > > Our SpikeSR outperforms ESTNet in both PSNR and SSIM. When compared with the strong baseline MambaIRv2, SpikeSR achieves slightly lower PSNR but delivers a higher SSIM, still demonstrating competitive performance.
> > >
> > > As noted in our limitations section, **the representation capacity of SNNs remains improvable**, making it challenging to substantially surpass large-capacity ANN-based models, e.g., MambaIRV2. Nevertheless, our work presents the first attempt to explore SNNs for SR tasks, which mitigates the performance gap by a large margin while maintaining significantly lower energy consumption. Overall, we believe this improvement is meaningful and acceptable!
> > >
> > > Both **MambaIR** and **MambaIRv2** will be properly cited and discussed in our revised manuscript. Our future work will focus on unleashing the potential of SNNs for better SR performance.
> > >
> > > **While this initial attempt may not have fully met your expectations, we truly appreciate that none of the other reviewers questioned its novelty. We believe this represents a meaningful and important first step in this direction. Even a one-point increase in your score would mean a great deal to us, and we sincerely hope you might consider it. Thank you very much for your support!**

---

> ### Author Response · Authors · 2025-08-05
> **Response to Official Comment by Reviewer DFR5**
>
> Dear Reviewer DFR5:
>
> Thank you again for your thoughtful and constructive comments. We have made our best effort to address your concerns in this round of discussion. We sincerely hope our responses resolve your doubts, and we would be truly grateful if you would consider raising your score — your recognition means a great deal to us.
>
> A1: **The differences:** We agree that the differences between natural and remote sensing images need clearer elaboration. Specifically, two key differences make SNNs suitable for remote sensing SR tasks:
> 1. Remote sensing images often cover significantly larger areas and have much higher resolutions. These characteristics pose more critical demands on efficient reconstruction of large-scale images, especially for on-orbit processing. By employing SNNs, which consume extremely low energy for computation, our method offers a new perspective on developing efficient models in large-scale Earth observation scenarios.
> 2. Remote sensing images present unique data properties such as scale diversity of objects and weaker texture details than natural images. In this case, we reveal a novel finding that SNNs maintain active neuro state across different scales of degradation, even when high-frequency textures are severely damaged. This insight suggests that SNNs are well-suited for remote sensing image SR tasks.
>
> A1: **To address these unique issues:** We have technically adapted the SNN architecture, making it more capable for remote sensing applications. In particular, our DSA exploits the rich self-similarity priors inherent in large-scale remote sensing scenarios to modulate the spatial-wise membrane potential, thus enabling faithful and high-fidelity reconstruction.
>
> A1: **The reason for comparing with natural image SR methods:** Currently, existing works of remote sensing SR were mainly adopted from or built upon natural image SR models. Following the common practice in the field, we chose widely recognized and representative natural SR for comparison. To address this concern, we further added ESTNet (TIP 2024) [R1], a lightweight model specifically developed for remote sensing images. We will try our best to complete the evaluation before the end of the discussion phase and ensure the results are included in the revised manuscript!
>
> [R1] Kang X, Duan P, Li J, Li S. Efficient Swin Transformer for Remote Sensing Image Super-Resolution. IEEE Transactions on Image Processing, 2024, 33: 6367-6379.
>
> We will incorporate these responses into the manuscript to better establish **the connection between SNNs and remote sensing!**
>
> A2: We appreciate the reviewer pointing out these important works. While our method is inspired by attention spiking neural networks, we would like to clarify that it exhibits significant differences in both motivation, architectural design, and experimental focus:
>
> 1. **Task-Specific Design for Super-Resolution:** Prior works [1–3] focus mainly on object classification or general vision tasks, whereas our method is specifically designed for the regression-based SR tasks, which pose specific challenges in recovering pixel-level details. This fundamental difference leads to distinct design choices in spike encoding, attention mechanism adaptation, and decoding structure.
>
> 2. **First to Explore SNN in Remote Sensing SR:** To our best knowledge, we are the first to explore the application of SNNs for remote sensing image SR, a scenario distinguished by unique challenges such as large-scale imaging, scale diversity, weak textures, etc.
>
> 3. **New Observations on Spiking Behavior under Degradation:** We reveal a novel empirical observation that SNN model maintains active neural states across diverse degradations, even when the high-frequency details are severely damaged. This behavior has not been reported in prior studies, and we believe it provides a new perspective on the potential of SNNs in handling degraded or low-quality visual inputs, particularly in large-scale Earth observation areas.
>
> We will discuss these instructive works carefully and add a detailed discussion in the revised manuscript!
>
> A3: We appreciate the valuable feedback. In response, we have added MambaIR v2 for comparison. We will do our best to report the results before the end of the discussion phase; otherwise, we assure that the updated comparisons will be included in the revised manuscript!
>
> Once again, thank you for your valuable time and thoughtful response!

---

> > ### Author Response · Authors · 2025-08-07
> > **Any further feedback?**
> >
> > Dear Reviewer,
> >
> > We hope the above replies can address your concerns and welcome any further feedback you may have!
> >
> > Authors

---

> > ### Author Response · Authors · 2025-08-08
> > **Any Additional Concerns?**
> >
> > Dear Reviewer DFR5:
> >
> > Many thanks for your time!
> >
> > We noticed there has been no response to our latest feedback, and would like to kindly confirm whether our updates have addressed your concerns.
> >
> > As the discussion phase is approaching its end, please be assured that we are ready to clarify any remaining questions you might have!
> >
> > Best regards,
> >
> > All authors

---

### Official Review · Reviewer_bD3C · 2025-07-03

**Clarity:** 3
**Significance:** 4
**Originality:** 3
**Rating:** 5
**Confidence:** 5

**Summary:**

The core content of this paper is to propose an efficient remote sensing image super-resolution (SR) method based on spiking neural networks (SNNs), named SpikeSR. This method optimizes the performance of SNNs in processing remote sensing image super-resolution tasks by combining the attention mechanism while maintaining high computational efficiency.

**Questions:**

Same weakness as above, see it.

**Ethical Concerns:**

["NO or VERY MINOR ethics concerns only"]

**Final Justification:**

Thanks to the author of the rebuttal. Judging from the supplementary efficiency comparison, while the proposed SpikeSR uses less energy, it does consume more GPU memory and inference time. Therefore, it is still an attempt to apply Spiking Neural Networks to remote sensing image super-resolution. I still think it is very innovative, but further research is needed.

Therefore, I maintain my original rating.

**Limitations:**

Same weakness as above, see for details.

**Paper Formatting Concerns:**

The main body of the paper has no obvious grammatical and expression errors. The paper has a clear structure and logical coherence, and can well convey the core content and contribution of the research.

**Quality:**

4

**Strengths And Weaknesses:**

### Strengths
1. For the first time, the spiking neural network (SNN) is combined with the attention mechanism for remote sensing image super-resolution (SR), and the spike attention block (SAB) and SpikeSR framework are proposed, filling the gap of SNN in the field of remote sensing SR.
HDA jointly optimizes spatiotemporal and channel features to solve the information loss problem caused by discrete pulses in SNN.
DSA uses the global self-similarity prior of remote sensing images to alleviate misaligned matching through deformable convolution and improve the authenticity of reconstruction.

2. The experimental design is sufficient, the multi-scenario verification covers 30 types of remote sensing scenes, the ablation experiment is sufficient, and the contribution of HDA/DSA is analyzed.

### Weaknesses
1. The paper emphasizes the "high energy efficiency" of SNN, but does not provide power consumption comparison data (such as comparison with ANN's Joule/inference), and only uses FLOPs to indirectly reflect efficiency. Can the author provide information such as inference speed and resource usage?
2. HDA relies on the external module TCJA, but does not explain how to adapt to the pulse characteristics of SNN (such as membrane potential dynamics), and the technical details are not complete.

---

> ### Author Rebuttal · Authors · 2025-07-28
>
> ## **Response to Reviewer bD3C**
>
> Dear Reviewer bD3C:
>
> We sincerely appreciate your time and effort in reviewing our paper. Thank you for your positive feedback on the significance of our work, especially recognizing our **first attempt** to combine spiking neural network (SNN) with attention mechanism for remote sensing image super-resolution. We are glad to hear that you find our SpikeSR **fills the gap of SNN in the field** of remote sensing SR, and that our **experimental design and ablation are sufficient**. We hope our responses below will address your concerns. We would greatly appreciate it if you would consider a score update.
>
> ### Weaknesses
> **W1: Can the author provide more information to reflect efficiency?**
>
>  A1: Thank you for this valuable comment. We have additionally evaluated three metrics to address this concern: (1) energy consumption (Joule/inference), (2) GPU memory usage, and (3) inference time. The results are presented in the table below. It can be found that our SpikeSR exhibits significantly lower energy consumption compared to ANN-based methods, which further demonstrates the **high energy efficiency of SNN**.  We will add a comprehensive discussion of model efficiency in our revised manuscript!
>
> | Methods | EDSR | CARN | SwinIR-light | OmniSR | NGswin | SPIN | HiT-SR | SpikeSR-S | SpikeSR-M | SpikeSR |
> | :---: | :---: | :---: | :---: | :---: | :---: | :---: | :---: | :---: | :---: | :---: |
> | Energy (pJ)  | 162.46 | 129.25 | 70.39 | 227.14 | 40.74 | 60.51 | 67.33 | **2.96** | 4.15 | 4.74 |
> | Memory (MB)  | 202.00 | 6.00 | 126.00 | 146.00 | 10.00 | **4.00** | **4.00** | 6.00 | 10.00 | 14.00 |
> | Times (s)  | 0.41 | **0.29** | 7.38 | 4.81 | 3.99 | 7.93 | 2.20 | 4.52 | 4.84 | 5.18 |
>
> It should be noted that the energy consumption is a theoretical approximation, calculated based on the following principle: the energy cost of ANNs is derived from multiply-and-accumulate (MAC) operations, while that of SNNs comes from AC operations (i.e., an addition). In a typical CMOS system [1], one MAC operation consumes approximately 3.2 pJ, while one accumulate AC consumes about 0.2 pJ. The energy consumption of an ANN model can be estimated by $E_{\text{ANN}} =  N_{\text{MAC}} \times E_{\text{MAC}}$, where $N_{\text{MAC}}$ is the number of MAC and $E_{\text{MAC}}$ is set to 3.2. For an SNN model, the energy consumption depends on the number of AC operations, the synaptic activity sparsity, and the number of time steps. The estimated energy is computed as $E_{\text{SNN}} =  s \times T \times N_{\text{AC}} \times E_{\text{AC}}$, where $s$ denote the mean sparsity, which is set to 16.42% [2], $T$ is the number of time setp, $N_{\text{AC}}$ means the additon mumber, and $E_{\text{AC}}$ is set to 0.2.
>
> The GPU consumption and inference time were evaluated on a single NVIDIA RTX 4090 GPU. The reported inference time is the total time required to super-resolve 100 LR images of size 160×160.
>
> **Reference**
>
> [1] Sze V, Chen Y H, Yang T J, et al. Efficient processing of deep neural networks: A tutorial and survey. Proceedings of the IEEE, 2017, 105(12): 2295-2329.
>
> [2] Guo Y, Chen Y, Liu X, et al. Ternary spike: Learning ternary spikes for spiking neural networks. Proceedings of the AAAI conference on artificial intelligence. 2024, 38(11): 12244-12252.
>
> **W2: Explain the adaptation and technical details of the proposed HDA.**
>
> Thank you for your insightful comment. To effectively adapt the HDA module to the spiking dynamics of SNNs, we first apply a squeeze operation to compress membrane potentials into an average matrix. We then perform 1D convolutions independently along the temporal and channel dimensions to extract attention weights that reflect the dynamic importance of features across both scopes. To further model their interdependence, we employ element-wise multiplication to fuse two weights into a unified temporal-channel attention map, which is ultimately used to modulate the membrane potential representations. Instead of operating directly on raw spike trains, HDA enhances feature representation by dynamically adjusting membrane potentials, enabling it to leverage rich spatiotemporal information while respecting the discrete nature of SNNs. This explanation will be included in the revised manuscript!
>
> Due to space limitations, we have provided more technical details of HDA in the supplementary materials, which is ensured to be publicly accessible. Besides, we will release the source code to ensure technical completeness and reproducibility.

---

> > ### Comment · Reviewer_bD3C · 2025-08-07
> >
> > Thanks to the author of the rebuttal. Judging from the supplementary efficiency comparison, while the proposed SpikeSR uses less energy, it does consume more video memory and inference time. Therefore, it is still an attempt to apply Spiking Neural  Networks to remote sensing image super-resolution. I still think it is very innovative, but further research is needed.
> >
> > Therefore, I maintain my original rating.

---

> > > ### Author Response · Authors · 2025-08-08
> > > **Official Comment by Authors**
> > >
> > > Dear Reviewer bD3C:
> > >
> > > Thank you for recognizing that our work is very innovative and for your thoughtful support throughout the review process. We truly appreciate your time and constructive feedback, and we wish you all the best!
> > >
> > > Best regards,
> > >
> > > Authors

---

> ### Author Response · Authors · 2025-08-05
>
> Dear reviewer,
>
> Thank you for the comments on our paper.
>
> We have submitted the responses to your comments. Please let us know if you have additional questions so that we can address them during the discussion period.
>
> Thank you!
>
> Best,
>
> Authors

---

### Comment · Area_Chair_ZW9c · 2025-08-07
**Please participate in discussions with authors**

Dear Reviewers,

I see that some of you have not yet responded to the author's rebuttal with further feedback. Please make sure to do so, which is important to reach a well thought out  consensus for the quality of this paper.

Best regards,
Submission16406 AC

---

> ### Author Response · Authors · 2025-08-07
>
> Dear Reviewers,
>
> Yes, we sincerely welcome any feedback you may have on our rebuttal and look forward to discussing it further with you!
>
> Authors

---

### Note · Authors · 2025-08-11

Dear Reviewers and AC:

We thank all reviewers and the AC for their valuable time, constructive feedback, and thoughtful discussions. We have carefully addressed all concerns raised during the rebuttal and discussion phases, and our final remarks are as follows:

1. We appreciate that, after the discussion, none of the reviewers questioned the novelty of our work; indeed, multiple reviewers explicitly recognized SpikeSR as **very innovative** (Reviewer bD3C), **a valuable and highly original contribution** (Reviewer WHSi), **a novel framework** (Reviewer DFR5), and **relatively novel** (Reviewer Encp). To the best of our knowledge, this is the first exploration to mitigate the research gap in SNN-based SR, providing a new perspective on developing efficient models in large-scale Earth observation scenarios.

2. In response to reviewer suggestions, we included a comprehensive analysis of model efficiency (memory, energy, and inference time) and performance comparisons against strong baselines such as **SeemoRe (ICML 2024), CATANet (CVPR 2025), ESTNet (TIP 2024), and MambaIRv2 (CVPR 2025)**. SpikeSR consistently demonstrates competitive performance while maintaining **significantly lower energy cost**.

3. While our current attempt has some limitations, e.g., the representation capacity remains improvable, we believe it marks a solid and promising first step in applying energy-efficient SNNs to challenging pixel-wise regression tasks, particularly for reconstructing large-scale, high-resolution remote sensing imagery.

We sincerely hope the AC will consider the novelty, significance, and potential impact of our work when making the final decision.

Thank you again for your time, support, and consideration!

All authors

---

### Decision · Program_Chairs · 2025-09-17

**Decision:**

Accept (poster)

**Comment:**

This paper proposes an efficient remote sensing image super-resolution (SR) method based on spiking neural networks (SNNs). It combines the attention mechanism while maintaining high computational efficiency. Experiment with 30 types of remote sensing scenes shows competitive results. Even though the result method is more energy efficient, its capacity and quality is still not as good as that of ANNs.

Reviewers recognize the potential in the direction of leveraging SNNs. However, there is still concern about lack of  theoretic justification to the design in connection to SNNs. Also more comparison to strong recent methods and SNN-based Transformers will make the work more complete.